

**The long-term trend and production sensitivity change of the U.S. ozone pollution from**
**observations and model simulations**
Hao He[1,2], Xin-Zhong Liang[1,2], Chao Sun[1], Zhining Tao[3,4], and Daniel Q. Tong[1,5]
[1]Department of Atmospheric and Oceanic Science, University of Maryland, College Park,
Maryland 20742, USA
[2]Earth System Science Interdisciplinary Center, University of Maryland, College Park, Maryland
20740, USA
[3]Universities Space Research Association, Columbia, Maryland 21046, USA
[4]NASA Goddard Space Flight Center, Greenbelt, Maryland 20771, USA
[5]Center for Spatial Information Science and Systems, George Mason University, Fairfax, VA
22030, USA
**Keywords:** Air Quality Trend, CMAQ Simulations, Ozone Production Sensitivity
Corresponding to Dr. Xin-Zhong Liang (xliang@umd.edu)





**Abstract**

19          We investigated the ozone pollution trend and its sensitivity to key precursors from 1990

to 2015 in the United States using long-term EPA AQS observations and mesoscale simulations.
The modeling system, a coupled regional climate – air quality (CWRF-CMAQ) model, well
captured summer surface ozone pollution during the past decades, having a mean slope of linear
regression with AQS observations at ~0.75. While the AQS network has limited spatial coverage
and measures only a few key chemical species, the CWRF-CMAQ provides comprehensive
simulations to enable a more rigorous study of the change in ozone pollution and chemical
sensitivity. Analysis of seasonal variations and diurnal cycle of ozone observations showed that
peak ozone concentrations in the summer afternoon decreased ubiquitously across the United
States, up to 0.5 ppbv/yr in major non-attainment areas such as Los Angeles, while
concentrations at other hours such as the early morning and late afternoon increased slightly.
Consistent with the AQS observations, CMAQ simulated a similar decreasing trend of peak
ozone concentrations in the afternoon, up to 0.4 ppbv/yr, and increasing ozone trends in the early
morning and late afternoon. While a monotonic decreasing trend (up to 0.5 ppbv/yr) in the odd
oxygen ($O_x = O_3 + NO_2$) concentrations are simulated by CMAQ at all daytime hours. This result
suggests that the increased ozone in the early morning and late afternoon was likely caused by
reduced $NO-O_3$ titration driven by continuous anthropogenic $NO_x$ emission reductions in the past
decades. Furthermore, the CMAQ simulations revealed a shift in chemical regimes of ozone
photochemical production. From 1990 to 2015, surface ozone production in some metropolitan
areas, such as Baltimore, has transited from VOC-sensitive environment (>50% probability) to
$NO_x$-sensitive regime. Our results demonstrated that the long-term CWRF-CMAQ simulations
can provide detailed information of the ozone chemistry evolution under a changing climate, and



may partially explain the U.S. ozone pollution responses to regional and national regulations.

## 1. Introduction

Tropospheric ozone ($O_3$) is one of the major air pollutants, regulated by the U.S.
Environmental Protection Agency (EPA), that pose myriad threats to public health and the
environment (Adams et al., 1989; WHO, 2003; Ashmore, 2005; Anderson, 2009; Jerrett et al.,
2009). It is also an important greenhouse gas due to the absorption of thermal radiation, affecting
the climate (Fishman et al., 1979; Ramanathan and Dickinson, 1979; IPCC, 2013). The major
source of tropospheric ozone is photochemical production from ozone precursors such as carbon
monoxide (CO), volatile organic compounds (VOCs), and nitrogen oxides ($NO_x$) at the presence
of sunlight (Crutzen, 1974; Seinfeld, 1991; Jacob, 2000; EPA, 2006), while downward transport
of stratospheric air mass contributes substantially to ozone concentrations in upper troposphere
(Levy et al., 1985; Holton et al., 1995; Stevenson et al., 2006). In the past decades, ozone
pollution in the United States has been reduced substantially due to regulations on anthropogenic
emissions of ozone precursors (Oltmans et al., 2006; Lefohn et al., 2008, 2010; Cooper et al.,
2012; He et al., 2013; Cooper et al., 2014), although some studies suggested no trend or slight
increases at some rural areas (Jaffe and Ray, 2007; Lefohn et al., 2010; Cooper et al., 2012).
Most of these analyses focused on peak ozone concentrations, e.g., daily maximum 8-hour
average ozone (MDA8), during summer, but studies on trends in seasonal and diurnal patterns of
ozone pollution are limited. He et al. (2019) analyzed measurements from four monitoring sites
in the eastern United States and found different ozone trends between rural and urban sites from
the late 1990s to the early 2010s including some increases at certain hours, suggesting effects of
national regulations could be regionally dependent. Thus, it is important to extend our study to



other regions of the United States in a longer time period.

The non-monotonic trends in United States ozone pollution could be caused by the

complex non-linear chemistry of ozone production involving $NO_x$ and VOCs (Logan et al., 1981;
Finlayson-Pitts and Pitts, 1999; Seinfeld and Pandis, 2006). With continuous reduction of
anthropogenic emissions of ozone precursors mainly $NO_x$ and VOCs in the United States, we
need to better understand the photochemical regime change for local ozone production (i.e.,
ozone production sensitivity), because air pollution regulations could have different effects under
$NO_x$-sensitive and VOC-sensitive environment (Dodge, 1987; Kleinman, 1994). For instance,
under a VOC-sensitive photochemical regime, the decrease of $NO_x$ emissions has limited
impacts on improving ozone pollution. Previous studies have developed photochemical
indicators to identify the ozone production sensitivity (Sillman, 1995; Sillman et al., 1997;
Tonnesen and Dennis, 2000b, a; Sillman and He, 2002). Sillman (1999) found the ratio of VOCs
and $NO_x$ ($VOC/NO_x$) has a typical value less than 4 for the VOC-sensitive environment and
higher than 15 for the $NO_x$-sensitive regime. Observation-based studies of ozone production
sensitivity relied on research grade measurements of ozone precursors and photochemical
intermediates that are not routinely measured by air quality management agencies such as the
U.S. EPA. These species include reactive nitrogen compounds ($NO_y$), nitric acid ($HNO_3$), and
hydrogen peroxide ($H_2O_2$), normally observed during field campaigns (e.g., Shon et al., 2007;
Peng et al., 2011) which only covered limited areas in certain periods. Studies based on air
quality models (AQM) could identify the ozone production regimes at regional scales (Sillman et
al., 1997; Sillman and He, 2002; Zhang et al., 2009a; Zhang et al., 2009b; Xie et al., 2011), but
the simulation periods were usually short (less than one year) and thus could not capture the
long-term change in ozone production sensitivity.





Regional AQMs are widely used for investigating the U.S. air quality (Tagaris et al.,
2007; Tang et al., 2009; Hogrefe et al., 2011; Pour-Biazar et al., 2011; He et al., 2016a; He et al.,
2018). They incorporate finer resolutions, more detailed emissions, and more explicit chemical
mechanism than global chemical transport models to better resolve characteristics of
tropospheric and surface dynamics, physical and chemical processes essential for air quality. Our
group has developed and used coupled regional climate-air quality models to study air quality
variations under a changing regional climate (Huang et al., 2007; Zhu and Liang, 2013; He et al.,
2016a; He et al., 2018). Our previous studies showed the model's ability to capture the decadal
U.S. air quality change (e.g., Zhu and Liang, 2013). In this study, we coupled he latest Climate-
Weather Research Forecast (CWRF) and the EPA Community Multiscale Air Quality (CMAQ)
models. CWRF has demonstrated substantial improvement in downscaling regional climate and
extremes (Liang et al., 2012; Chen et al., 2016; Liu et al., 2016a; Liang et al., 2019a; Sun and
Liang, 2019a; Sun and Liang, 2019b) and thus can provide more realistic weather conditions for
AQMs to produce more credible air quality simulations.
To supplement the limited observations in both spatial coverage and chemical species, we
conducted a continuous 26-yr CWRF-CMAQ simulation from 1990 to 2015 for a more rigorous
analysis of long-term U.S. ozone trend. The model performance of the U.S. air quality was first
evaluated against gridded ozone observations. The ozone seasonal variations and diurnal cycles
were then extracted to determine the observed long-term trend. The model simulations were
subsequently analyzed to explain the observed ozone trends and change in ozone production
sensitivity.





## 2. Observations and model simulations

### 2.1 Long-term EPA observations

Hourly measurements of surface ozone concentrations from 1990 to 2015 were available from the EPA Air Quality System (AQS) database (https://www.epa.gov/outdoor-air-quality-data). They have been examined following the EPA guidance including the quality assurance and quality control. The locations and durations of AQS monitoring sites have changed substantially due to logistics and requirements to cover the regions sensitive to air pollution. Figure 1 shows that more than 2000 sites reported ozone measurements from 1990 to 2015. To alleviate the impacts from missing data and short durations, we selected 640 sites that had ozone observation records longer than 20 years. Hourly ozone observations were processed following the approach described in He et al. (2019) to create the long-term seasonal and diurnal records for these stations.

### 2.2 Regional climate modeling

CWRF (Liang et al., 2012) was driven by the European Centre for Medium-Range Weather Forecasts ERA-Interim reanalysis (ERI, Dee et al., 2011) to downscale regional climate variations during 1989-2015 with the first year as the spin-up and not used. We adopted the well-established North American domain with a 30-km grid spacing (Fig. 1), covering the Contiguous United States (CONUS) and neighboring southern Canada, northern Mexico and adjacent oceans. The CWRF incorporated advanced representations of key physical processes and integrations of external forcings crucial to climate scales (Liang et al., 2012). It has been vigorously tested in North America and Asia showing outstanding performance to capture regional climate characteristics (Yuan and Liang, 2011; Qiao and Liang, 2015; Chen et al., 2016;





Liu et al., 2016b; Qiao and Liang, 2016; Liang et al., 2019b). The CWRF downscaling has been
shown to provide realistic meteorological fields and regional climate signals that can be cordially
used to drive the CMAQ for long air quality simulations. Major CWRF physics configurations
include the semi-empirical cloudiness parameterization of Xu and Randall (1996), the cloud
microphysics scheme of Tao et al. (1989), the short wave and long wave radiation scheme of
Chou et al. (2001), the ensemble cumulus parameterization (Qiao and Liang, 2015, 2016; Qiao
and Liang, 2017), and the planetary boundary layer scheme of Holtslag and Boville (1993).
Hourly CWRF outputs were processed using a modified Meteorology-Chemistry Interface
Processor (MCIP, version 4.3) for CMAQ simulations.

**2.3 Emissions preparation**

To prepare anthropogenic emissions, we chose 2014 as the baseline year. This year's

emissions were modified from the National Emissions Inventory 2011 (NEI2011). The
modifications was based on measurements from the Ozone Monitoring Instrument (OMI)
onboard satellite Aura, the ground-based AQS network, and the *in-situ* continuous emissions
monitoring in power plants (Tong et al., 2015; Tong et al., 2016). The so modified NEI2011
inventory was processed using the Sparse Matrix Operator Kernel Emissions (SMOKE) version
3.7 (Houyoux et al., 2000). Emissions from on-road, off-road, and area sources were placed at
the model layer closest to the surface. Emissions from point sources, e.g., stacks from power
plants, were distributed vertically based on stack height and plume rise. The plume rise was
estimated based on the method in Briggs (1972). The inventory pollutants were speciated
according to the carbon bond chemical mechanism version 5 (CB05) and AERO5 aerosol
mechanism. To fill the gap where NEI2011 data were not available, the Emissions Database for



Global Atmospheric Research (EDGAR v3, http://edgar.jrc.ec.europa.eu/) at a 1° × 1° resolution
developed by the Joint Research Centre of European Commission was adapted. Figure 2 shows
an example of 2010-2015 mean $NO_x$ emissions distribution over the modeling domain. Daily
mean $NO_x$ emissions have high values in urban areas of cities such as Los Angeles, Chicago, and
the northeast corridor from Washington D.C. to Boston.

To project emissions from the baseline year into all individual years, we used the scaling

factors from Air Pollutant Emissions Trends Data compiled by the U.S. EPA
(https://www.epa.gov/air-emissions-inventories/air-pollutant-emissions-trends-data).   Figure   3
shows the emission evolution from 1990 to 2015. Since 1990 anthropogenic emissions of $NO_x$,
CO, sulfur dioxide ($SO_2$), and VOCs had steady decreasing trends, with $SO_2$ experiencing the
largest reduction. On the other hand, anthropogenic $PM_{2.5}$ and $NH_3$ emissions stayed mostly flat
since the early 2000s.
The wildfire emissions were based on the Global Fire Emissions Database, Version 4 with
small fires (GFEDv4s, Randerson et al., 2017; van der Werf et al., 2017). The 0.25° × 0.25°
degree resolution GFEDv4s data were projected onto the modeling domain and speciated into the
CB05 and AERO5 species. GFEDv4s had a monthly resolution from 1997 to 2000 and daily
resolution from 2000 onward. Figure 4 illustrates the fire emissions evolution during 1990 to
2015 relative to 2014. Fire emissions have large interannual variations, with high emissions in
1998, 2002, 2013, and 2015, and low emissions in 2001, 2004, and 2014. We developed a
method to merge the aforementioned anthropogenic and wildfire emissions into the
temporalized, gridded and speciated data ready for CMAQ.
The biogenic emissions were calculated online within CMAQ based on the Biogenic
Emissions Landuse Database, Version 3 (BELD3, https://www.epa.gov/air-emissions-



[modeling/biogenic-emissions-landuse-database-version-3-beld3](). The 1-km resolution BELD3
data with spatial distribution of 230 vegetation classes over the North America were processed
through the Spatial Allocator developed by the Community Modeling and Analysis System
(CMAS) center ([https://www.cmascenter.org/sa-tools/]) to generate the gridded vegetation
distribution over the study domain. Table 1 lists the 5-yr mean variations of daily major ozone
precursor (CO, $NO_x$, and NMVOCs) emissions in the modeling domain and five subdomains.
The emission data show regionally dependent reductions. For instance, compared with 2000-
2004, the $NO_x$ emissions in 2005-2009 decreased by ~36% averaged in the CONUS, while 38%
and 35% reductions existed in states of California and Texas.

**2.4 Air quality modeling**
The EPA CMAQ model version 5.2 (EPA, 2017) was selected to simulate the U.S. air
quality variations driven by CWRF meteorological fields (Section 2.2) and constructed emissions
(Section 2.3). Major chemical mechanisms include the Carbon Bond 6 revision 3 (CB6r3) gas
phase chemical scheme with updated secondary organic aerosol (SOA) and nitrate chemistry
(Yarwood et al., 2010) and the latest AERO6 aerosol scheme (EPA, 2017), which improved U.S.
air quality simulations over previous chemical mechanisms (Appel et al., 2016). Chemical initial
and boundary conditions were obtained from the default concentration profiles built in CMAQ
(EPA, 2017). Simulations were conducted continuously for each 5-year segment (e.g., 1990-
1994, 1995-1999, etc.) with two-week spin-up in December prior to each starting years to speed
up simulation turn around. Hourly concentrations of ozone and its key precursors such as nitric
oxide (NO) and nitrogen dioxide ($NO_2$) were saved for subsequent analyses.





## 3. Results

### 3.1 Evaluation of CMAQ performance

Our previous studies showed that the direct comparison of observation data from monitoring sites and CMAQ results in 30-km grid could introduce inconsistency for evaluating the model performance (He et al., 2016a). So we applied the EPA Remote Sensing Information Gateway (RSIG) software (available at https://www.epa.gov/rsig) to map the site observations onto our CMAQ grid. Figure 5 compares summer (JJA) mean MDA8 ozone in 2014 between gridded AQS observations and CMAQ outputs and shows that the model can well capture the U.S. ozone pollution, except underestimation in urban areas such as the Los Angeles basin.

Table 2 summarized the statistics for CMAQ performance of the summer ozone concentrations during 2000 - 2015 in CONUS and subdomains. Linear regression analyses of MDA8 ozone result in a mean slope value of 0.75 for CONUS, i.e., CMAQ slightly underestimates ozone over the United States. In subdomains, CMAQ performance exhibits large interannual variations. For instance, in Texas the linear regression slope and correlation coefficient ranges from 0.58 to 0.97 and 0.55 to 0.86, respectively. Generally, this modeling system has substantially improved performance in the Southeast, California and Texas, and moderately improved performance in the Northeast and Midwest as compared with our previous study (He et al., 2016a). These results demonstrate the ability of CWRF-CMAQ to credibly simulate historical air quality.

### 3.2 Long-term ozone trend in AQS observations

We applied a box-averaging technique (He et al., 2016b; He et al., 2019) to analyze ozone measurements at the selected AQS monitoring sites (Fig 1). This approach used an hour by





month box to calculate the mean 24-hr diurnal cycle of ozone for each month. Then we
calculated the climatology mean over 24 hours by 12 months and the respective anomaly for
each month at each AQS site. Figure 6 shows samples of long-term mean ozone concentrations
and anomalies at four non-attainment cities: Baltimore, Maryland; Los Angeles, California;
Denver, Colorado; and New York City (NYC), New York. The hour by month climatology (left
column of Fig. 6) shows that the peak ozone concentrations in the afternoon during the ozone
season (April to September) have been reduced significantly in these cities. However, ozone
concentrations in the morning (8 am to 12 pm, all local time hereafter) and at night (8 pm to 8
am) increased slightly. These results confirm the effectiveness of recent emission controls which
were designed to reduce the peak ozone. But the expansion of ozone at moderate levels (40-50
ppbv), which are higher than the natural background of U.S. ozone (Fiore et al., 2002; Fiore et
al., 2003; Wang et al., 2009; Lefohn et al., 2014), could cause negative health impacts.

The anomaly (right column of Fig. 6) shows large variabilities of ozone concentrations

because the ozone production is significantly impacted by regional climate (e.g., temperature,
precipitation) with interannual and decadal variations. Large ozone reduction occurred after 2003
when the EPA $NO_x$ State Implementation (SIP) call was implemented (He et al., 2013). The
anomalies at Los Angeles (Fig. 6b) and NYC (Fig. 6d) shows decreases of the peak ozone in the
afternoon of summer and increases in other times and seasons. For Baltimore and Denver, the
peak ozone was not monotonically reduced, but increased in some years after 2002. Given the
continuous reduction of anthropogenic emissions in the past decades, the increased ozone
pollution in these areas could be caused by other factors such as higher summer temperatures  in
certain years or enhanced stratosphere-troposphere exchange (for Denver at the high altitude
area), which need further investigations in the future.



We used the linear regression analysis to calculate the slope, correlation (R), and p-value

of ozone trend at each local hour. Figure 7 shows ozone trends (slope, unit of ppbv/yr) at AQS
sites which are statistically significant ($R^2 > 0.5$, and $p < 0.05$) in the early morning (8 am), at
noon (12 pm), in the afternoon (4 pm), and in the evening (8 pm). Consistent results with the
four cities (Fig. 6) are found ubiquitously. The peak ozone at noon and in the afternoon generally
had a decreasing trend in CONUS, up to 0.5 ppbv/yr, confirming the improved air quality due to
regulations, while ozone in the early morning and late afternoon increased slightly at most of
monitoring sites. However, AQS sites in the Bay area (San Francisco, California) and Denver
had stronger positive trends in the day time. The possible explanations include the trans-pacific
transport of ozone and its precursors to the U.S. West Coast (Hudman et al., 2004; Huang et al.,
2010; Lin et al., 2012b) and stratosphere-troposphere exchange of ozone to high altitude region
(Langford et al., 2009; Lin et al., 2012a).

**3.3. Ozone trends derived from CMAQ simulations**

We applied the same box-averaging technique to hourly surface ozone simulations

in CONUS and conducted the linear regression analysis to estimate the ozone trend at each
model grid (Fig. 8). Compared with ozone trends derived from AQS observations (Fig. 7), the
CMAQ model successfully captured the spatial pattern and magnitude of change in ozone
pollution. For instance, at 4 pm LT, CMAQ simulated up to 0.4 ppbv/yr decrease in surface
ozone in the eastern United States and south region of California state. However, CMAQ
simulated statistically insignificant trends (white color in Fig. 8c) at 4 pm LT in the Bay area,
Los Angeles, and Denver where AQS observations showed increasing trends (Fig. 7c). The
discrepancy occurred because our model used the static chemical lateral conditions (LBCs) that



did not include the change of trans-Pacific transport of air pollutants, which were known to
elevate the background ozone in the West Coast. Also CMAQ does not contain stratospheric
chemistry and hence cannot account the contribution of downward transport of stratospheric
ozone to the high altitude region.

Consistent with trends derived from AQS observations, CMAQ also simulated increasing

ozone trends in the early morning (8 am LT, Fig. 8a) and late afternoon (8 pm LT, Fig 8d),
especially in urban regions such as Los Angeles and Chicago. He et al. (2019) found ozone
increases from observations at four sites in the eastern United States and a possible cause
suggested by the reduced NO-$O_3$ titration through examining the trend in odd oxygen ($O_x = O_3 +$
$NO_2$). Due to known interferences from nitrogen compounds such as $NO_x$ and organic nitrates to
standard $NO_2$ measurements employed by EPA (Fehsenfeld et al., 1987; Dunlea et al., 2007), the
analysis of $O_x$ required research grade $NO_2$ analyzer (e.g., photolytic $NO_2$ conversion) which are
not available in current AQS network. Thus, our simulations provide a unique opportunity to
expand such study to the whole CONUS.

Trends in $O_x$ concentrations simulated by CMAQ at 8 am, 12 pm, 4 pm, and 8 pm show a

consistent decreasing trend over the modeling domain, up to 0.5 ppbv/yr reductions in the eastern
United States (Fig. 9). The result confirms our hypothesis that the reduced NO-$O_3$ titration
elevated surface ozone concentrations in the early morning and late afternoon when the
photochemical production of ozone is low or not active. The current EPA ozone standard focuses
on peak ozone concentrations, i.e., MDA8 ozone which usually has maximum values at noon or
in the early afternoon, so the damage from additional ozone exposure from these elevated ozone
concentrations in the early morning and late afternoon is not considered under the current
environment policy. These increased ozone levels could offset the benefit from reduced peak



ozone in past decades, which needs further investigation to provide scientific evidence for future
policy decision.

**294    3.4 Change in photochemical regime**

With the continuous reduction of ozone precursor emissions, changes in the complex $O_3$-

$NO_x$-VOC chemistry are anticipated. We used the $O_3/NO_y$ ratio as the indicator to study the
photochemical regime change in the U.S. surface ozone production. The threshold of 15
proposed by Zhang et al. (2009b) was adopted to identify the VOC-sensitive or $NO_x$-sensitive
regime, i.e., $O_3/NO_y < 15$ indicating the VOC-sensitive regime. For each local hour, we
calculated the probability when $O_3/NO_y$ is lower than 15 in every month. Figure 10 shows the
probability of VOC-sensitive regime at 2 pm in July of 1995, 2005, and 2015. Most regions
dominated by the VOC-sensitive chemistry are urban or suburban where anthropogenic $NO_x$
emissions are relatively high as compared with anthropogenic and/or biogenic VOCs emissions,
such as the Los Angeles basin, the Northeast corridor (Washington D.C.-Baltimore-Philadelphia-
NYC), and the Chicago metropolitan area. Noting that these maps are created based on ozone
photochemical production simulated at the surface level, so the distributions are slightly different
from recent studies using satellite data (Duncan et al., 2010; Jin et al., 2017; Ring et al., 2018).

We calculated the mean probability of VOC-sensitivity (2 pm in July) in a $3 \times 3$ grid in

metropolitan areas of Baltimore, Los Angeles, and NYC from 1990 to 2015 (Fig. 11). CMAQ
simulations suggest the transition from VOC-sensitive regime to $NO_x$-sensitive regime in these
urban areas. There were interannual variabilities in the probability of VOC-sensitive
photochemistry in Baltimore (~50%) and NYC (~80%) in the 1990s and the early 2000s. After
the EPA 2003 $NO_x$ SIP call, anthropogenic $NO_x$ emissions decreased substantially leading to



reduced ozone pollution in the eastern United States (He et al., 2013), so the photochemical
production of surface ozone is expected to gradually become $NO_x$-sensitive. In 2015, ozone
photochemical production in Baltimore was dominated by $NO_x$ emissions (only ~20%
probability of VOC-sensitive), while NYC had higher probability (>50%) of VOC-sensitive
chemistry. In Los Angeles, ozone chemistry slowly leaned to $NO_x$-sensitive, but until 2015 the
local ozone production was still controlled by VOCs emissions. In regions with VOC-sensitive
photochemistry in summer, reduction in $NO_x$ emissions had a limited impact on the local rate of
ozone production until the photochemistry of ozone production became $NO_x$-sensitive. Our
analysis can partially explain the different responses of ozone pollution in major U.S. cities to
national air quality regulations during the past decades (Cooper et al., 2012) and can provide
some insights for future policy decision.

## 4. Conclusions and Discussion

EPA AQS observations in the United States from 1990 to 2015 were analyzed to study the

trend in surface ozone seasonal variations and diurnal cycles. We showed that the peak ozone
concentrations in the afternoon decreased significantly, especially in major non-attainment
regions, but the concentrations in the early morning and late afternoon increased slightly.
Regional climate-air quality model captured the long-term records of U.S. ozone pollution and
suggested that the increased ozone was caused by reduced $NO-O_3$ titration due to the continuous
reduction of $NO_x$ emissions. Model simulations also showed changes in ozone photochemical
regime. The U.S. urban/suburban areas generally transited from the VOC-sensitive regime in the
early 1990s to more $NO_x$-sensitive regime recently. But ozone production in some cities such as
NYC and Los Angeles are still substantially impacted by VOC emissions. The current national





and regional regulations focus on the MDA8 ozone concentrations mainly determined by the
peak ozone in the afternoon. Our study revealed the elevated ozone concentrations in the early
morning and late afternoon which must be considered for their impacts on public health. While
$NO_x$ emissions are currently the main target of national and regional control measures, our study
suggested that regulations on anthropogenic VOCs emissions could be important in certain
regions. This study can improve our understanding about the effectiveness of regulations in the
past decades and will provide scientific evidence for future policy decision.

Ozone production is highly non-linear, so accurate emissions are essential to simulate its

long-term variations. Due to limited resources, we scaled the anthropogenic emissions from a
baseline year (2014) to the 1990s using factors derived from the national trend data. This scaling
cannot accurately reflect the detailed regional-dependent regulations for individual state such as
the 2012 Health Air Act in Maryland (He et al., 2016b). Also, because the GFED data were only
available after 1997, the contribution of wildfire emissions to ozone pollution was not included
in model simulations between 1990 to 1996. Thus, we anticipated some uncertainties in ozone
simulations in the early 1990s. Our model also has limitations to reproduce ozone records in high
altitude regions such as Denver because of lacking the stratospheric chemistry in CMAQ and
missing the effect of stratosphere-troposphere exchange to surface ozone. Lastly, due to limited
resources, our experiments used static chemical LBCs for CMAQ, which excluded the long-
range transport of air pollutants into the United States. For some West Coast regions such as the
state of California, the trans-Pacific transport had been enhanced in the past decades and could
play a more important role in determining the local air quality. To accurate evaluate the
contribution from trans-boundary emission, dynamic LBCs from a global chemical transport
model is needed in the future study.

**Author contribution**

H.H., X.L., and Z.T. designed the experiment; H.H. and C.S. developed the CWRF-CMAQ

system and performed the CWRF modeling; Z.T. and D.T. prepared the emission data; H.H.

conducted the CMAQ simulations; H.H., Z.T., and C.S. analyzed the data; H.H. prepared the

manuscript with contributions from all co-authors.

**Acknowledgments**

This work was supported by the U.S. Environmental Protection Agency under Assistance

Agreement No. RD-83587601. It has not been formally reviewed by EPA. The views expressed

in this document are solely those of the authors and do not necessarily reflect those of the

funding Agency. EPA does not endorse any products or commercial services mentioned in this

publication. We thank the support of University of Illinois at Urbana-Champaign

(UIUC)/USEPA award 20110150701. We thank the National Center for Supercomputing

Applications (NCSA) and the National Center for Atmospheric Research (NCAR) Computation

and Information System Laboratory for supercomputing support. We thank Dr. Plessel Todd for

the help on the RSIG software (https://www.epa.gov/rsig).

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



**Tables and Figures**

**Table 1.** Summary of multiyear mean average of daily CO, $NO_x$, and NMVOCs emissions in the CONUS and five subdomains. (Unit: mol/km$^2$ per second) Please note that our California and Texas subdomains include more area than the states of California and Texas.

| CONUS | | | | | Southeast | | |
|---|---|---|---|---|---|---|---|
| Year | CO | $NO_x$ | NMVOCs | | CO | $NO_x$ | NMVOCs |
| 1990-1994 | 32.9 | 1.24 | 0.94 | | 47.2 | 1.43 | 1.03 |
| 1995-1999 | 26.2 | 1.18 | 0.76 | | 37.4 | 1.36 | 0.85 |
| 2000-2004 | 18.9 | 1.26 | 0.69 | | 26.4 | 1.46 | 0.72 |
| 2005-2009 | 12.3 | 0.94 | 0.60 | | 16.9 | 1.07 | 0.59 |
| 2010-2015 | 8.0 | 0.60 | 0.46 | | 11.0 | 0.66 | 0.45 |
| California | | | | | Northeast | | |
| 1990-1994 | 18.3 | 1.22 | 0.57 | | 110.3 | 3.29 | 2.12 |
| 1995-1999 | 14.6 | 1.16 | 0.46 | | 87.2 | 3.16 | 1.68 |
| 2000-2004 | 10.6 | 1.23 | 0.40 | | 62.1 | 3.41 | 1.43 |
| 2005-2009 | 7.1 | 0.91 | 0.35 | | 40.3 | 2.56 | 1.25 |
| 2010-2015 | 4.6 | 0.56 | 0.26 | | 25.9 | 1.62 | 0.93 |
| Texas | | | | | Midwest | | |
| 1990-1994 | 22.6 | 1.21 | 1.26 | | 58.2 | 1.88 | 1.41 |
| 1995-1999 | 18.1 | 1.15 | 1.03 | | 46.3 | 1.80 | 1.14 |
| 2000-2004 | 13.0 | 1.20 | 1.01 | | 33.4 | 1.92 | 0.98 |
| 2005-2009 | 8.4 | 0.91 | 0.92 | | 22.0 | 1.44 | 0.85 |
| 2010-2015 | 5.5 | 0.60 | 0.73 | | 14.3 | 0.91 | 0.63 |





**Table 2.** Summary about the comparison of JJA MDA8 ozone concentrations from AQS
observations and CMAQ simulations during 2000-2015 in the CONUS and subdomains. Slope
and Correlation (Corr. R) are calculated for each year based on linear regression analysis. Please
note that our California and Texas subdomains include more area than the states of California
and Texas.

| Year | Slope | Corr. R | NMB | RMSE | Year | Slope | Corr. R | NMB | RMSE |
|---|---|---|---|---|---|---|---|---|---|
| **CONUS** | | | | | | | | | |
| **2000** | 0.73 | 0.37 | -6.9 | 10.5 | **2008** | 0.70 | 0.54 | -5.4 | 8.4 |
| **2001** | 0.80 | 0.61 | -7.7 | 8.7 | **2009** | 0.78 | 0.35 | -1.6 | 8.5 |
| **2002** | 0.71 | 0.63 | -8.6 | 9.2 | **2010** | 0.75 | 0.51 | -6.2 | 8.4 |
| **2003** | 0.81 | 0.60 | -4.3 | 8.4 | **2011** | 0.77 | 0.42 | -7.1 | 9.2 |
| **2004** | 0.85 | 0.39 | 1.3 | 8.9 | **2012** | 0.67 | 0.60 | -10.7 | 9.3 |
| **2005** | 0.87 | 0.54 | -7.3 | 8.8 | **2013** | 0.70 | 0.50 | -1.8 | 7.9 |
| **2006** | 0.77 | 0.48 | -7.6 | 9.1 | **2014** | 0.72 | 0.44 | -3.0 | 7.6 |
| **2007** | 0.70 | 0.60 | -6.1 | 8.0 | **2015** | 0.73 | 0.41 | -4.2 | 7.7 |
| **California** | | | | | | | | | |
| **2000** | 0.70 | 0.67 | -19.3 | 15.2 | **2008** | 0.63 | 0.53 | -18.0 | 14.8 |
| **2001** | 0.72 | 0.63 | -18.1 | 14.8 | **2009** | 0.67 | 0.61 | -19.0 | 13.5 |
| **2002** | 0.80 | 0.55 | -15.5 | 14.4 | **2010** | 0.62 | 0.55 | -19.0 | 14.1 |
| **2003** | 0.80 | 0.55 | -20.1 | 16.2 | **2011** | 0.68 | 0.57 | -17.0 | 13.3 |
| **2004** | 0.78 | 0.51 | -19.2 | 16.1 | **2012** | 0.64 | 0.63 | -21.4 | 14.9 |
| **2005** | 0.78 | 0.54 | -19.0 | 15.3 | **2013** | 0.64 | 0.60 | -17.9 | 13.5 |
| **2006** | 0.80 | 0.61 | -20.5 | 15.6 | **2014** | 0.69 | 0.56 | -21.9 | 14.8 |
| **2007** | 0.69 | 0.65 | -16.0 | 12.9 | **2015** | 0.72 | 0.61 | -22.3 | 14.2 |
| **Texas** | | | | | | | | | |
| **2000** | 0.60 | 0.77 | -20.4 | 11.8 | **2008** | 0.62 | 0.74 | -10.5 | 6.6 |
| **2001** | 0.58 | 0.62 | -19.6 | 11.5 | **2009** | 0.73 | 0.78 | -17.1 | 8.7 |
| **2002** | 0.70 | 0.72 | -10.4 | 6.6 | **2010** | 0.65 | 0.77 | -9.4 | 5.3 |
| **2003** | 0.64 | 0.78 | -8.8 | 6.5 | **2011** | 0.52 | 0.83 | -22.7 | 12.1 |
| **2004** | 0.97 | 0.55 | -7.2 | 5.8 | **2012** | 0.53 | 0.86 | -17.8 | 9.4 |
| **2005** | 0.70 | 0.78 | -21.5 | 11.4 | **2013** | 0.53 | 0.74 | -11.6 | 6.9 |
| **2006** | 0.66 | 0.83 | -20.5 | 11.3 | **2014** | 0.66 | 0.72 | -5.0 | 4.7 |
| **2007** | 0.77 | 0.84 | -4.0 | 3.9 | **2015** | 0.76 | 0.61 | -10.1 | 5.8 |
| **Southeast** | | | | | | | | | |
| **2000** | 0.61 | 0.41 | -20.5 | 13.3 | **2008** | 0.52 | 0.77 | -13.4 | 8.3 |
| **2001** | 0.64 | 0.70 | -7.7 | 6.2 | **2009** | 0.88 | 0.52 | -2.7 | 4.2 |



| 2002 | 0.56 | 0.77 | -14.1 | 9.5 | 2010 | 0.69 | 0.75 | -7.8 | 5.1 |
|---|---|---|---|---|---|---|---|---|---|
| 2003 | 0.65 | 0.77 | -0.7 | 4.7 | 2011 | 0.84 | 0.62 | -13.5 | 8.2 |
| 2004 | 0.81 | 0.59 | 3.2 | 4.4 | 2012 | 0.62 | 0.73 | -9.4 | 6.1 |
| 2005 | 0.54 | 0.64 | -8.8 | 6 | 2013 | 0.74 | 0.70 | 7.0 | 4.1 |
| 2006 | 0.74 | 0.60 | -14 | 9 | 2014 | 0.84 | 0.40 | 0.9 | 4.0 |
| 2007 | 0.56 | 0.71 | -14.1 | 9 | 2015 | 0.71 | 0.44 | -2.6 | 4.2 |
| **Northeast** | | | | | | | | | |
| 2000 | 0.50 | 0.25 | 7.9 | 7.0 | 2008 | 0.46 | 0.11 | -0.5 | 5.8 |
| 2001 | 0.46 | 0.28 | -3.6 | 6.0 | 2009 | 0.67 | 0.23 | 13.7 | 7.3 |
| 2002 | 0.51 | 0.13 | -8.5 | 8.3 | 2010 | 0.49 | 0.10 | -0.4 | 5.6 |
| 2003 | 0.85 | 0.16 | 3.0 | 5.3 | 2011 | 0.47 | 0.31 | 3.2 | 5.9 |
| 2004 | 0.81 | 0.21 | 10.0 | 6.6 | 2012 | 0.55 | 0.17 | -2.9 | 5.3 |
| 2005 | 0.84 | 0.11 | 2.5 | 5.8 | 2013 | 0.78 | 0.45 | 11.6 | 6.4 |
| 2006 | 0.45 | 0.21 | 3.0 | 6.0 | 2014 | 0.60 | 0.33 | -4.8 | 5.1 |
| 2007 | 0.48 | 0.19 | -0.7 | 5.6 | 2015 | 0.49 | 0.11 | 2.2 | 5.1 |
| **Midwest** | | | | | | | | | |
| 2000 | 0.41 | 0.25 | 3.4 | 5.9 | 2008 | 0.44 | 0.25 | 3.5 | 4.7 |
| 2001 | 0.55 | 0.30 | -2.3 | 4.9 | 2009 | 0.54 | 0.22 | 14 | 7.2 |
| 2002 | 0.45 | 0.27 | -5.2 | 7.0 | 2010 | 0.57 | 0.12 | 2.4 | 5.3 |
| 2003 | 0.66 | 0.25 | -0.1 | 4.7 | 2011 | 0.45 | 0.21 | 1.1 | 5.6 |
| 2004 | 0.68 | 0.44 | 13.9 | 7.5 | 2012 | 0.46 | 0.19 | -11.6 | 8.3 |
| 2005 | 0.76 | 0.15 | -4.4 | 5.6 | 2013 | 0.74 | 0.18 | 4.9 | 4.0 |
| 2006 | 0.50 | 0.17 | 0.3 | 5.0 | 2014 | 0.64 | 0.20 | 5.7 | 4.1 |
| 2007 | 0.39 | 0.20 | -0.6 | 5.6 | 2015 | 0.68 | 0.27 | 8.7 | 4.7 |

NMB: Normalized Mean Bias (Unit: %)
RMSE: Root Mean Square Error (Unit: ppbv)



**Figure 1.** Locations of EPA AQS sites for surface ozone monitoring during 1990-2015. Red dots stand for monitoring sites with more than 20 year record. Black dots show the locations of monitoring sites have short data records which are not used in this study. The map shows the CWRF-CMAQ 30-km domain and five subdomains sensitive to air pollution. CA: California (including nearby parts of Nevada, Arizona and Oregon); TX: Texas (including nearby parts of Louisiana, Arkansas, and Oklahoma); SE: Southeast; NE: Northeast; MW: Midwest. Please note that our CA and TX subdomains include more area than the states of California and Texas.

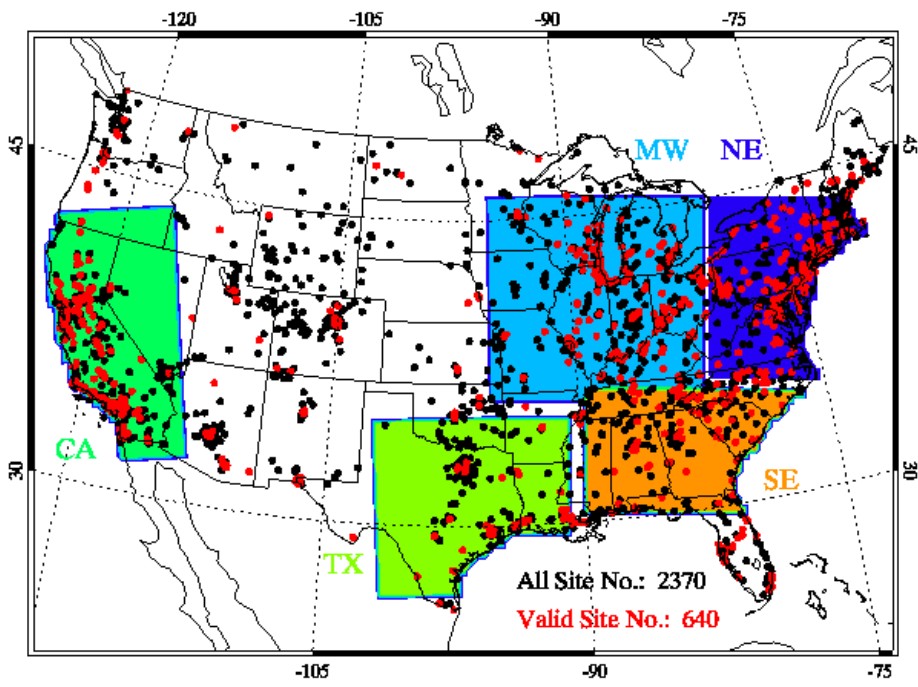




**Figure 2.** Averaged daily NO$_x$ emissions between 2010 and 2015 in the modeling domain (Unit:
mol/km$^2$ per second).

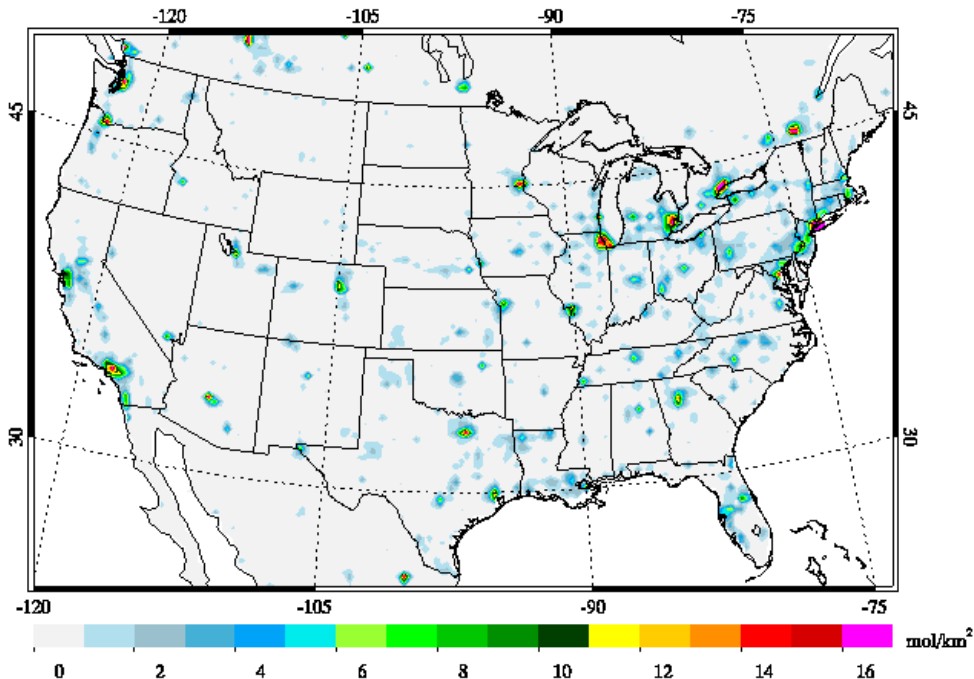




**Figure 3.** Anthropogenic emission evolution relative to 2014 in the modeling domain from 1990
– 2015.

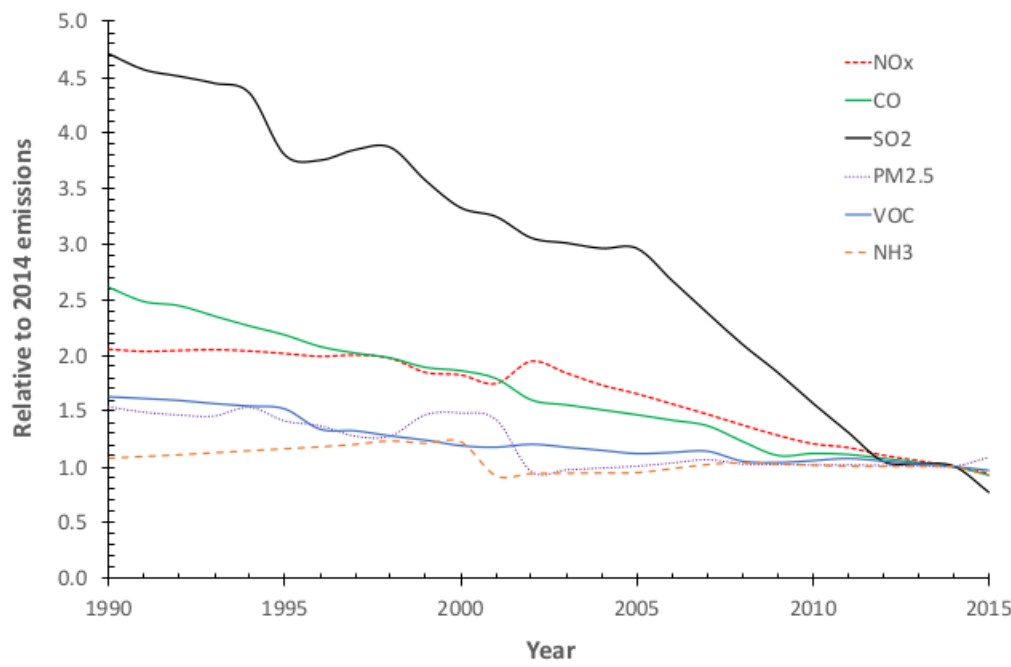







**Figure 4.** Fire emission evolution relative to 2014 in the modeling domain from 1990 – 2015.
Noting that GFED fire emissions are not available before 1997.

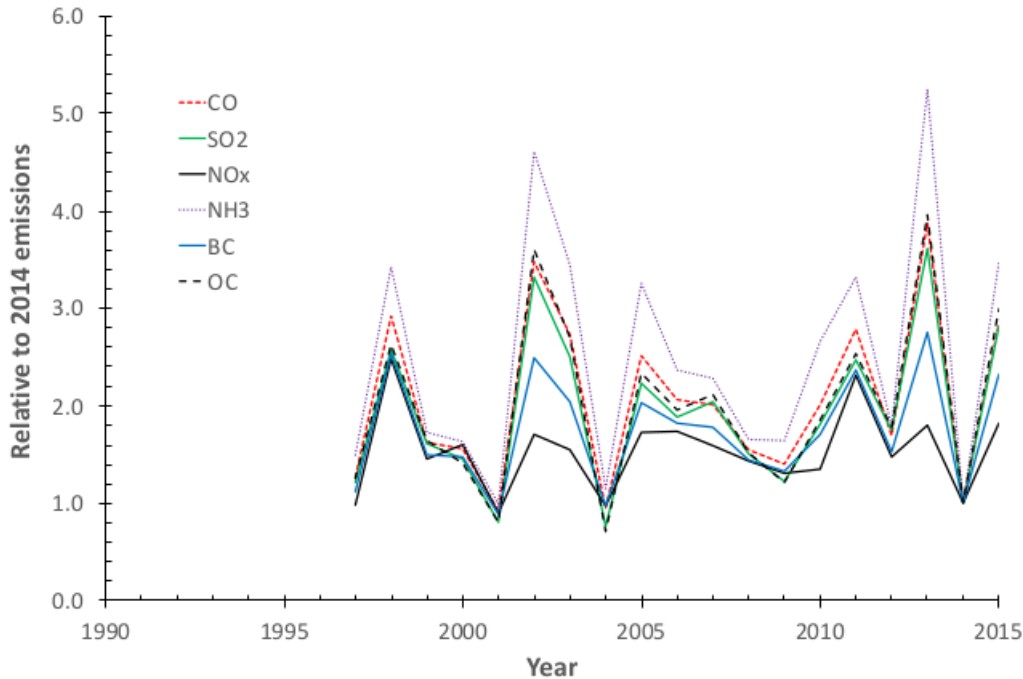






**Figure 5.** Comparison of summer MDA8 ozone concentrations from EPA AQS observations and
CMAQ simulations in 2014. AQS station data were gridded to the CMAQ grid using the EPA
RSIG software. a) Contour plot, the background stands for the CMAQ outputs and the dots stand
for gridded AQS observations; b) Scatter plot of the gridded AQS observations and co-located
CMAQ outputs.
a)

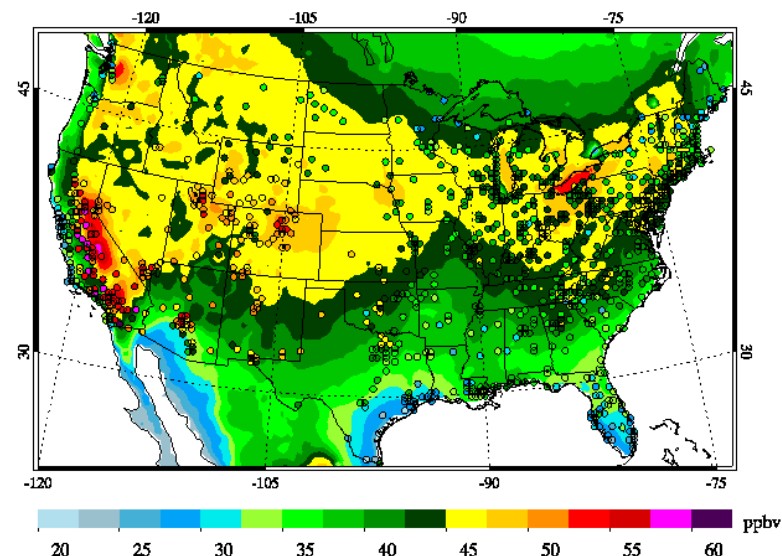


b)

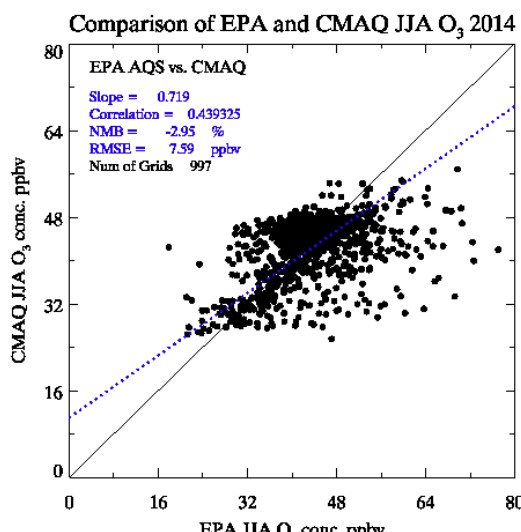






**Figure 6.** The box-averaging analyses of AQS ozone observations at selected sites from 1990-
2015. a) Essex, Maryland (suburban Baltimore, AQS ID 240053001); b) Pasadena, California
(downtown Los Angeles, AQS ID 060372005); c) Denver, Colorado (downtown Denver, AQS ID
080310014); d) Staten Island, New York (suburban New York City, AQS ID: 360850067). Left
column shows the monthly mean, right column shows the anomaly values. White patches stand
for missing data or not sufficient data for the box-averaging analysis.






**Figure 7.** Trend in ozone observations at selected EPA AQS sites during 1990-2015 (Unit:
ppbv/yr). a) at 8 am; b) at 12 pm; c) at 4 pm; d) at 8 pm (all local time). We only show the sites
with statistically significant linear trend in the plots.

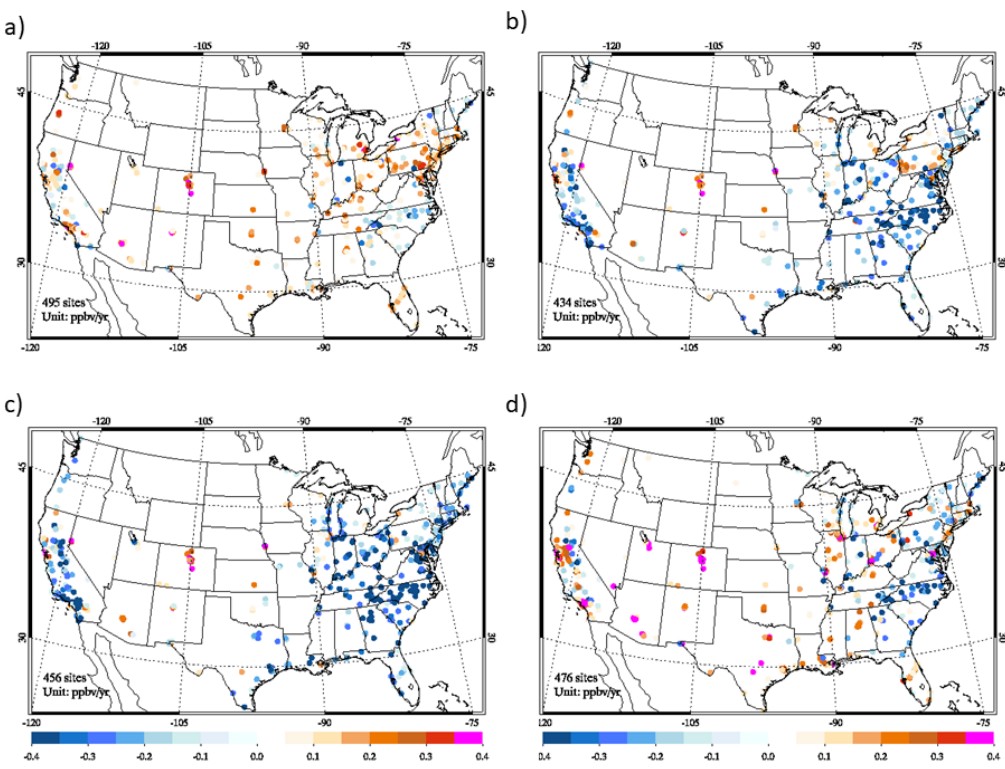




**Figure 8.** Trends in ozone simulations from CMAQ during 1990-2015 (Unit: ppbv/yr). a) at 8 am; b) at 12 pm; c) at 4 pm; d) at 8 pm (all local time). We only show CMAQ grids with statistically significant linear trend in the plots.

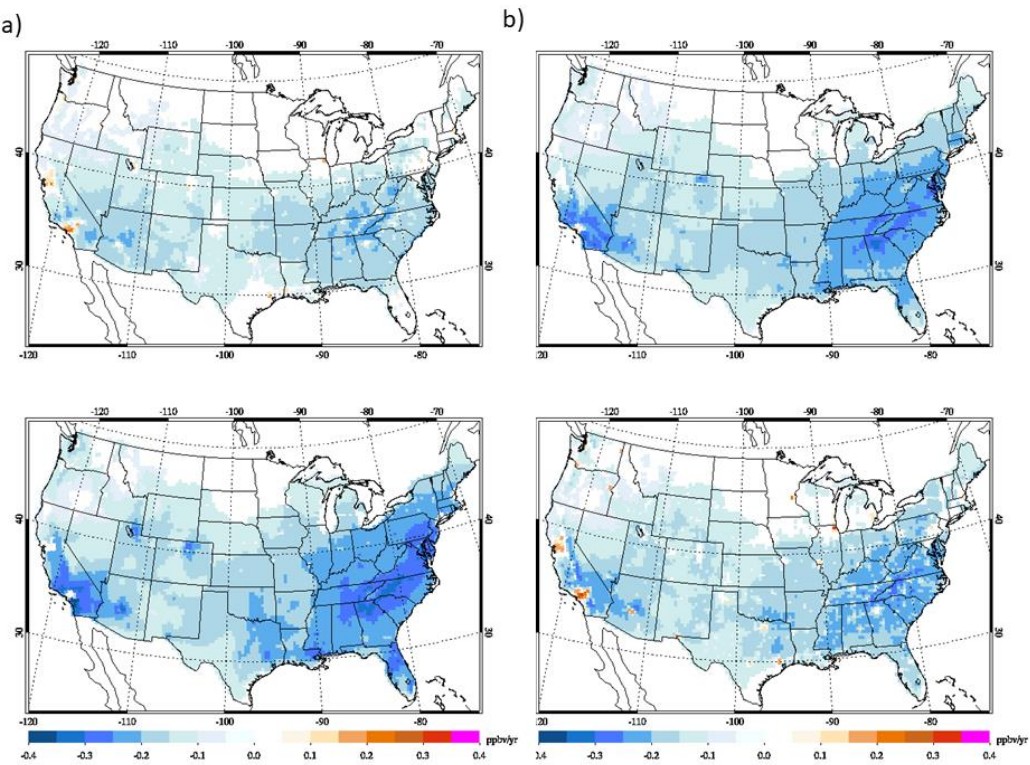



**Figure 9.** Trend in O$_x$ simulated by CMAQ during 1990-2015. a) at 8 am; b) at 12 am; c) at 4
pm; d) at 8 pm (all local time). We only show CMAQ grids with statistically significant linear
trend in the plots.

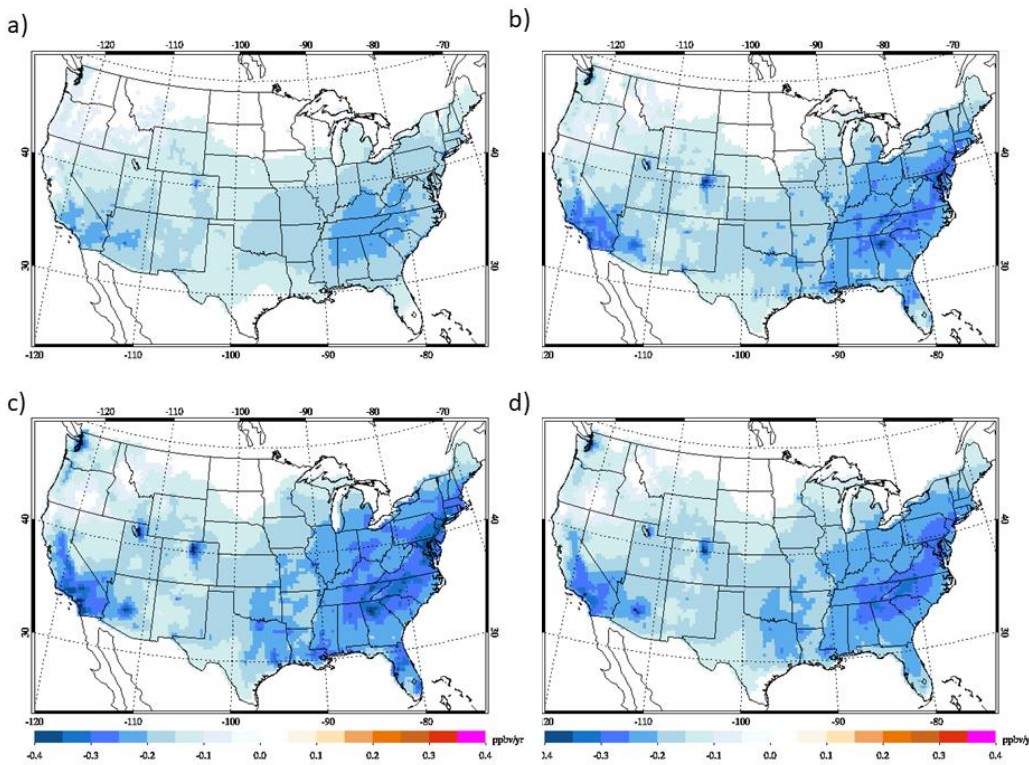






**Figure 10.** Probability of VOC-sensitive photochemical ozone production (i.e., $O_3/NO_y < 15$) in
the CONUS simulated by CMAQ at 2 pm local time in July, a) 1995; b) 2005; and c) 2015
a)

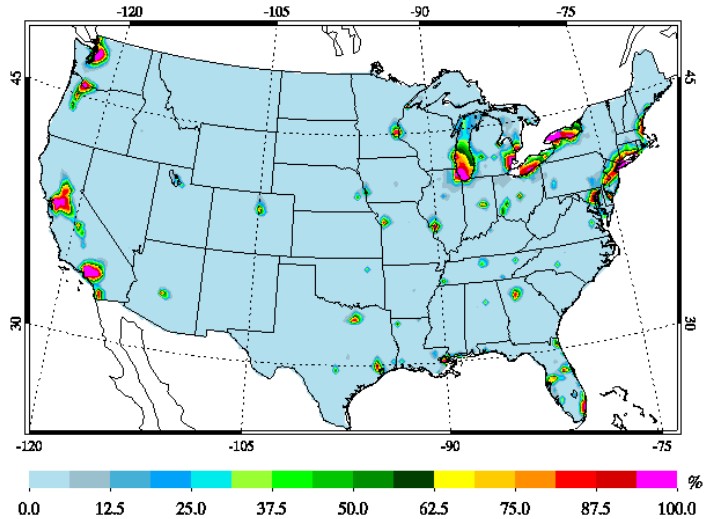



b)

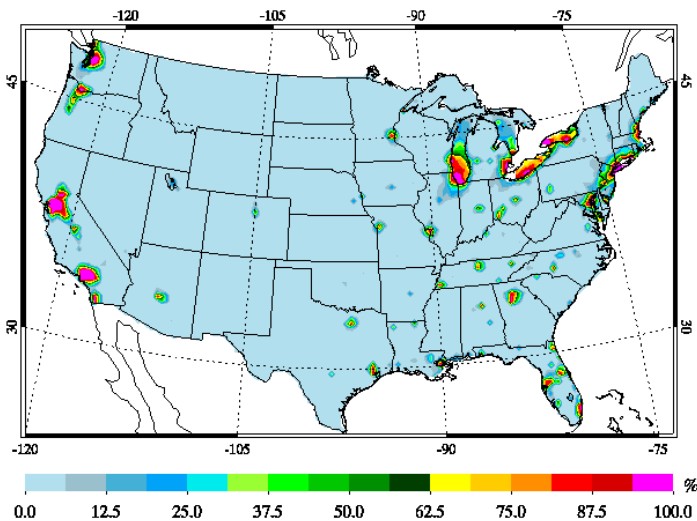





c)

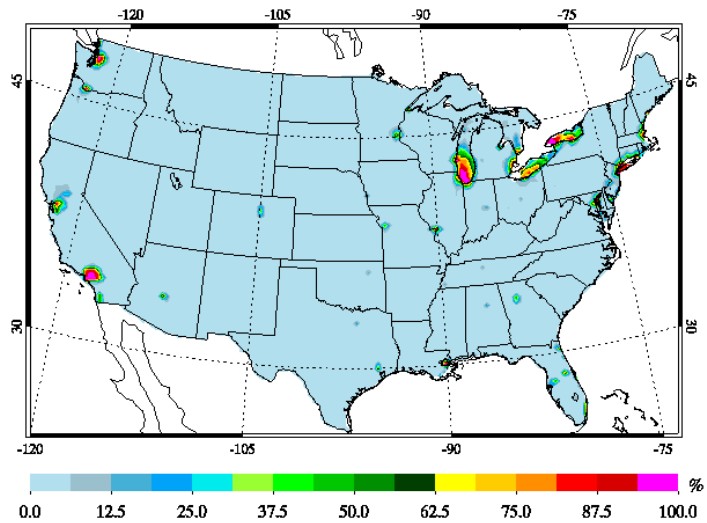






**Figure 11.** Long-term trends in probability of VOC-sensitive photochemical production of
surface ozone in three major urban areas at 2 pm in July. Probability is calculated using averages
of 3 × 3 grids centered at downtown.

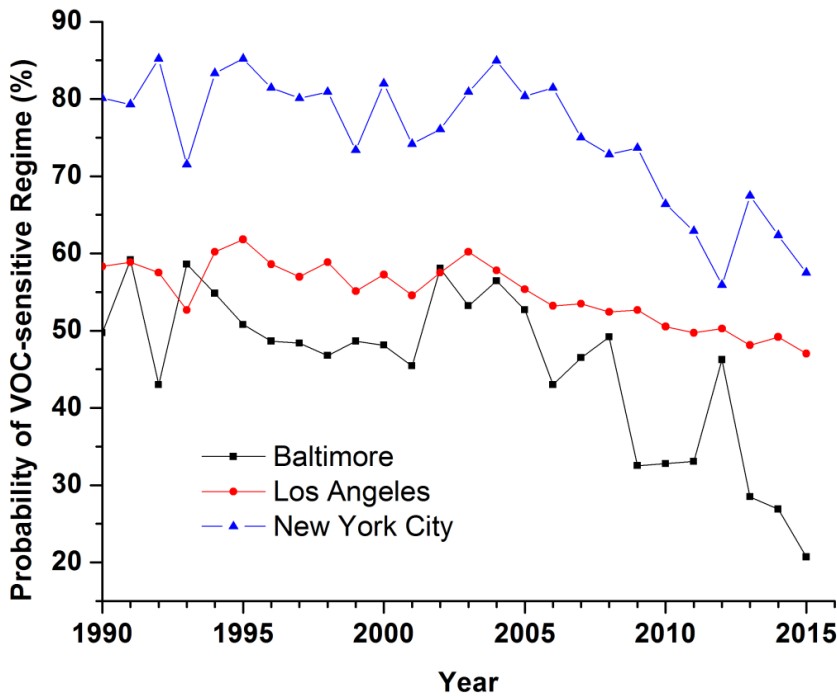
