# Peer review of "The long-term trend and production sensitivity change of the U.S. ozone pollution from"

_Atmospheric Chemistry and Physics, 2019_

## Referee Comment (RC1) · Anonymous Referee #1 · 12 Oct 2019

The authors presented a study on ozone pollution trend and its sensitivity to key precursors in the US over 1990 – 2015. While the lack of measurement data of ozone precursors in time and space makes it difficult to study long-term trends in ozone chemistry, increasingly available model simulations may be in place for such analysis with caution exercised. This study represents such effort. Studies like this one are needed for information of changes in ozone chemistry due to climate change and anthropogenic emissions control. I have a few comments as follows.

The authors stated that, to avoid introducing inconsistency for model evaluation caused by "direct comparison", they used the EPA RSIG software to visualize the observed

[Figure]

and modeled ozone values. The question I have is how they quantified the difference between the modeled and observed values besides directly comparing the modeled value from a grid and the observed value(s) contained in that grid. Incidentally, what did "gridded" mean in "AQS station data were gridded to the CMAQ grid" in the Figure 5 caption? Didn't they just superimpose the station data on the CMAQ simulated distribution there?

The authors stated that in subdomains CMAQ performance exhibited large interannual variations (Table 2) and they further stated that their CWRF-CMAQ simulations showed improved performance in the Northeast and Midwest. It would be illuminating to the modeling community if they could expand on those two statements by explaining why.

The authors brought up an interesting point that for Baltimore and Denver the peak ozone increased in some years after 2002 although anthropogenic emissions were continuously decreasing in the past decades. From there they inferred that the increased ozone pollution on those areas "could be caused by other factors such as higher summer temperatures in certain years or enhanced stratosphere-troposphere exchange" (lines 234-244). Their Figure 4 showed the largest peak of fires emissions over a couple of years starting in 2002, which could have influenced ozone during those years. Also, were the summer temperatures really higher and STE enhanced during those couple of years? The authors might want to avoid making such sheer speculations if they had no intention to get into making these points.

I am a bit concerned with their threshold value of the O3/NOy ratio used to examine changes in the ozone formation regime. In our experience the model simulated O3/NOy ratio could differ greatly from the observed values and between simulations using different models. I am not sure if the threshold value of 15 from Zhang et al. (2009b), which used completely different models and had very different emissions and chemical environments, would be applicable. The authors need to find their own threshold value for their model simulations.

---

## Referee Comment (RC2) · Anonymous Referee #3 · 1 Jan 2020

This manuscript presents a modeling study of the decadal ozone trend in the US. I am impressed by the significance of the results, but there are still several important issues need to be addressed before publication.

1 How will the results of CWRF-CMAQ differ from WRF-CMAQ? You can also use WRF to simulate decadal climate with long-term reanalysis. Better to include some description of the advantage of CWRF over WRF.

2 Scaling factors were used to get historical emissions. However, this will keep the spatial distribution the same at 2011 level. Why don't you use the information from other historical inventories, such as EDGAR? Could you discuss how this will affect the

results?

3 Chemical initial and boundary conditions were obtained from the default concentration profiles built in CMAQ. For long lived chemical species like ozone, long range transport and stratospheric intrusions would be important. If default concentration profiles are set, how to consider the historical changes in sources outside the US?

4 $O_3/NO_y$ ratio was used as the indicator of VOC or NOx limited. The threshold of was adopted ($O_3/NO_y < 15$ indicating the VOC-sensitive regime). How to demonstrate this threshold and ratio is proper and accurate or represent the sensitivity. As model usually has difficulty in capturing the concentrations of $NO_y$, the results might be questionable with this assumption.

---

## Author Comment (AC1) · 12 Feb 2020

Response to Reviewer #1's comments on He et al. 2019 Atmospheric Physics and Chemistry manuscript

We thank the anonymous reviewer for thoroughly reading our manuscript and providing helpful comments and suggestions, which lead a significant improvement of our manuscript. The detailed responses to major point comments are listed below (text in *italic* and black is the reviewer's comments, and the normal text highlighted in blue is our response):

*The authors presented a study on ozone pollution trend and its sensitivity to key precursors in the US over 1990 – 2015. While the lack of measurement data of ozone precursors in time and space makes it difficult to study long-term trends in ozone chemistry, increasingly available model simulations may be in place for such analysis with caution exercised. This study represents such effort. Studies like this one are needed for information of changes in ozone chemistry due to climate change and anthropogenic emissions control. I have a few comments as follows.*

Response: We appreciate the positive comments from the anonymous reviewer, and the manuscript has been revised according to these comments as listed below.
*Line numbers are based on the revised clean version of the manuscript.

*The authors stated that, to avoid introducing inconsistency for model evaluation caused by "direct comparison", they used the EPA RSIG software to visualize the observed and modeled ozone values. The question I have is how they quantified the difference between the modeled and observed values besides directly comparing the modeled value from a grid and the observed value(s) contained in that grid. Incidentally, what did "gridded" mean in "AQS station data were gridded to the CMAQ grid" in the Figure 5 caption? Didn't they just superimpose the station data on the CMAQ simulated distribution there?*

Response: The direct comparison is usually conducted through sampling the grid of CMAQ where the AQS site is located. In our previous study (He et al., 2016), we found that due to the uneven distribution of AQS monitoring sites, the direct comparison of AQS observations and our 30-km CMAQ simulations could be of problem. Figure 1 adapted from He et al. (2016) presents the case study of the CMAQ evaluation over California. AQS monitoring sites are point measurements and usually concentrated in populous urban and suburban areas such as the Los Angeles basin where high ozone levels prevail, but sparse in rural areas where ozone concentrations are generally low. Therefore, sampling CMAQ grids over locations of these AQS sites could introduce important biases. At that time, we lacked the capability to process large amount of AQS observations to our modeling grid (i.e., regrid observations to model grids), so we only raised this question. Recently, EPA added the capability in the Remote Sensing Information Gateway (RSIG) system (https://www.epa.gov/hesc/remote-sensing-information-gateway), which can calculate the gridded values of air pollutants on a selected model grid. The RSIG software applied the inverse-distance-weighted method to calculate the gridded mean (https://www.epa.gov/hesc/how-rsig-regrids-data), which is not a simple arithmetic mean of AQS observations within the grid. To explain the problem of direct comparison and the unique characteristics and advantage of RSIG, we added the following sentence in Line 215 as "*The direct comparison is usually conducted through sampling the grid of CMAQ where the AQS site*

*is located, while the distribution of AQS monitoring sites is usually uneven with more sites concentrated in populous urban and suburban areas where high ozone levels prevail. Sampling 30-km CMAQ grids over the locations of AQS measurements, i.e. direct comparison of averaged concentrations in the 900 km² CMAQ grid and pointwise AQS observations, could introduce important biases"* and in Line 222 as *"The RISG has the capability to 're-grid' the AQS observations on a selected model grid using the inverse-distance-weighted method to calculate the gridded mean concentrations (https://www.epa.gov/hesc/how-rsig-regrids-data)."*

[Figure]

Figure 1. Comparison of EPA AQS ozone observations (color dots) and model simulations (background) in California (adapted from He et al. (2016)).

*The authors stated that in subdomains CMAQ performance exhibited large interannual variations (Table 2) and they further stated that their CWRF-CMAQ simulations showed improved performance in the Northeast and Midwest. It would be illuminating to the modeling community if they could expand on those two statements by explaining why.*

Response: We appreciate that the reviewer raised this question. Previous studies usually aggregate and average data from all modeling years into one analysis without considering interannual variations. Figure 2 shows a comparison using 2000-2015 data for the CONUS, which has better performance (NMB = -1.3%) than the year-by-year evaluation summarized in Table 2. In our manuscript, we conducted the evaluation by year for the CONUS and five subdomains, so we can identify the years with good and bad performance. With the assumption that our emissions reflect the gradual reduction of anthropogenic emissions in the past decades, the year-to-year fluctuations of model performance should be related to climate signals that control the regional ozone pollution. By doing so, we effectively reduced the impact of emissions reduction on the model performance. We are investigating the relationships between regional

climate characteristics and ozone pollution especially extreme pollution episodes, and hope to better address your question from the perspective of how extreme events affect interannual variations.

About the model improvement, we apologize that the current manuscript is not stated very clearly. These improvements were achieved through comparison with our previous study employing the CMM5-CMAQv4.7 modeling system (He et al. 2016). CMM5 is the previous generation of regional climate model, developed on the MM5 model. Our CWRF model has shown better performance for downscaling the U.S. climate and the updated CMAQ v5.2 has also improved. We have demonstrated that CWRF with more sophisticated physical processes (Liang et al., 2012) can simulate better regional climate variations in the United State including surface temperature and precipitation (Chen et al., 2016; Liu et al., 2016), especially for extreme events (Sun and Liang, 2020a; Sun and Liang, 2020b). These meteorological variables are key factors for better air quality simulations. To make these two points clear, we added following sentences and revised this paragraph in Line 233 as "*With gradual reduction in anthropogenic emissions, the fluctuations of CMAQ performance could be related to climate signals which control the regional ozone pollution. Future work is needed to identify the relationship between these regional climate variations and the U.S. ozone pollution*" and in Line 236 as "*Generally, this modeling system has substantially improved performance in the Southeast, California and Texas, and moderately improved performance in the Northeast and Midwest as compared with our previous modeling system (He et al., 2016), which significantly underestimated the U.S. ozone pollution. One reason is that CWRF with more sophisticated representation of physical processes have the capability to better simulate the U.S. climate especially surface temperature and precipitation (Liang et al., 2012; Chen et al., 2016; Liu et al., 2016; Sun and Liang, 2020b; Sun and Liang, 2020a), which are key to accurate air quality simulations. The evaluation of CMAQ performance demonstrates the capability of CWRF-CMAQ to credibly simulate historical air quality.*"

[Figure]

Figure 2. Similar as Figure 5b in the main article, but with data from 2000-2015

*The authors brought up an interesting point that for Baltimore and Denver the peak ozone increased in some years after 2002 although anthropogenic emissions were continuously decreasing in the past decades. From there they inferred that the increased ozone pollution on those areas "could be caused by other factors such as higher summer temperatures in certain years or enhanced stratosphere-troposphere exchange" (lines 234-244). Their Figure 4 showed the largest peak of fires emissions over a couple of years starting in 2002, which could have influenced ozone during those years. Also, were the summer temperatures really higher and STE enhanced during those couple of years? The authors might want to avoid making such sheer speculations if they had no intention to get into making these points.*

Response: Thanks for pointing out that the wildfire emissions could be an important source for the increasing summer ozone in some regions, especially Denver which could be impacted by the wildfire activities in the western United States. Another paper under review (Tao et al., *Remote Sensing*, 2020) confirmed the impacts from wildfires on air quality in the western United States. Another possibility is the change of ozone production regime, especially in Baltimore as discussed in the later section. We checked the temperature anomaly at Essex MD (AQS ID: 240053001, Fig. 3), which did not support our hypothesis. We agree that these speculations about possible high regional temperatures and STE lack evidences in current study, and removed these hypotheses in the revised manuscript.

[Figure]

Figure 3. The AQS temperature anomaly at Essex, Maryland (Site ID: 240053001).

*I am a bit concerned with their threshold value of the O3/NOy ratio used to examine changes in the ozone formation regime. In our experience the model simulated O3/NOy ratio could differ greatly from the observed values and between simulations using different models. I am not sure if the threshold value of 15 from Zhang et al.(2009b), which used completely different models and had very different emissions and chemical environments, would be applicable. The authors need to find their own threshold value for their model simulations.*

Response: Thanks for this important question. We understand that the model simulated $O_3/NO_y$ ratio could differ largely from the observed values. In our study, we could not access the long record of research-grade $NO_y$ observations from the EPA network and did not conduct long-term sensitivity experiments of CMAQ with reduced emissions rates due to limited computation resources. So we have to rely on results from the previous studies. Sillman explored the concept using photochemical indicators including $O_3/NO_y$ to identify the regime of ozone photochemical production, finding that the link between the ozone production sensitivity and these indicators is largely unaffected by changes in model assumptions, including emission rates of anthropogenic and biogenic species (Sillman, 1995; Sillman et al., 1997). Observations from urban areas of Atlanta, New York, and Los Angeles was compared with modeling results from the Urban Airshed Model at urban scales, and a threshold of 7 was proposed for using $O_3/NO_y$ ratios as the photochemical indictor (Sillman et al., 1997). Zhang et al. (2009) expanded the study to the CONUS with 1-year CMAQ simulations, suggesting a threshold of 15 for $O_3/NO_y$ ratios. Zhang et al. (2009) used previous CMAQ version 4.4 for 1-yr CONUS simulations of 2001 at a relatively coarse spatial resolution (36 km) which is close to our 30-km CONUS domain, so we adopted their proposed threshold. We agree that the current manuscript lacked the evaluation of this threshold with our modeling system, and we developed the following approach to test it.

We selected hourly $O_3$, $NO_y$, and $NO_x$ concentrations from CMAQ in the afternoon (defined as 12 pm to 4 pm) in 2014, and calculated the $O_3/NO_y$ ratios. Figure 4a shows scatter density of $O_3/NO_y$ ratios vs. $NO_x$ concentrations, which is calculated based on a $100 \times 100$ bins with $NO_x$ from 0-20 ppbv $NO_x$ (i.e., 0.2 ppbv per bin) and 0-100 $O_3/NO_y$ ratios (i.e., 1 per bin). In the afternoon over the CONUS, the ozone production is mainly in high $O_3/NO_y$ ratio (>15) and low $NO_x$ (less than 2 ppbv) environment, i.e., in the $NO_x$-sensitive regions by thresholds proposed by both Sillman et al. (1997) and Zhang et al. (2009). Figure 4b shows the same density plot, but the color stands for mean $O_3$ concentrations. Both low and high ozone concentrations exist in high $NO_x$ region ($NO_x$ > 4 ppbv), which are usually urban or suburban. Then we calculated the weighted ozone concentrations that equals to the product of $O_3/NO_y$ and $NO_x$ scatter density (Fig. 4a) and mean $O_3$ concentrations (Fig. 4b), which stands for the $O_3$ sensitivity with respect to $O_3/NO_y$ ratios and $NO_y$ concentrations over the CONUS (Fig. 4c). At the national scale, when the weighted ozone concentrations increase with CMAQ $NO_x$ levels, the photochemical production is $NO_x$-sensitive. The region with $O_3/NO_y$ higher than 7 and 11 both have this characteristics, while due to low probability (Fig. 4a) and urban environment (Fig. 4b) we believe the $O_3/NO_y$ threshold of 7 stands for the urban environment. Thus, the $O_3/NO_y$ ratio threshold of 15 is more proper for the CONUS scale analysis. This analysis qualitatively supports our application of results from Zhang et al. (2009).

In summary, due to limited resources and experiment design, identifying a threshold of $O_3/NO_y$ ratio is beyond the scope of this study. Using results from our CMAQ model, we proved that the threshold of 15 should be practical for our study. We added the following sentences in Line 323 as "*The usage of O₃/NOy ratio was first proposed by Sillman (Sillman, 1995; Sillman et al.,*

*1997). Sillman et al. (Sillman et al., 1997) conducted a case study of observations in urban areas (Atlanta, New York, and Los Angeles) and modeling results from the Urban Airshed Model and suggested the threshold of 7 as the transition region from VOC-sensitive environment to $NO_x$-sensitive environment. Zhang et al. (2009a; 2009b) expanded this method to the CONUS with 1-year observations and CMAQ simulations (36-km spatial resolution) and suggested a threshold of 15 for ozone pollution at the national scale. In this study, we did not have access to the long-term research grade $NO_y$ observations from the AQS network and did not conduct sensitivity experiments (due to computational resource limit) with reduced $NO_x$ emissions following Sillman et al. (1997), so we have to reply on the $O_3/NO_y$ threshold from literature. We conducted a simple evaluation of our CMAQ results and found the threshold of 7 could be more proper for urban areas and the threshold of 15 should be more applicable for our study of the whole United State (Figure S1 in the supplementary material). Please note that the $O_3/NO_y$ ratio could depend on the modeling framework, so due to the similarity of our modeling system (30-km CMAQ) and the model used in Zhang et al. (2009a; 2009b), our analysis suggests the similar threshold of 15"*
and the discussion above to the supplementary material.

a)

[Figure]

b)

[Figure]

c)

[Figure]

Figure 4. Afternoon $O_3$/$NO_y$ ratios vs. $NO_x$ concentrations simulated by CMAQ in 2014. a) Scatter density, the color contour stands for the probability for each bin; b) $O_3$ concentrations, the color contour stands for the mean $O_3$ over the bins; c) Weighted $O_3$ concentrations. Two black lines stand for the $O_3$/$NO_y$ ratios of 7 and 11.

Reference:

Chen, L. G., Liang, X. Z., DeWitt, D., Samel, A. N., and Wang, J. X. L.: Simulation of seasonal US precipitation and temperature by the nested CWRF-ECHAM system, Climate Dynamics, 46, 879-896, 10.1007/s00382-015-2619-9, 2016.

He, H., Liang, X.-Z., Lei, H., and Wuebbles, D. J.: Future U.S. ozone projections dependence on regional emissions, climate change, long-range transport and differences in modeling design, Atmospheric Environment, 128, 124-133, https://doi.org/10.1016/j.atmosenv.2015.12.064, 2016.

Liang, X.-Z., Xu, M., Yuan, X., Ling, T., Choi, H. I., Zhang, F., Chen, L., Liu, S., Su, S., Qiao, F., He, Y., Wang, J. X. L., Kunkel, K. E., Gao, W., Joseph, E., Morris, V., Yu, T.-W., Dudhia, J., and Michalakes, J.: Regional Climate-Weather Research and Forecasting Model, Bulletin of the American Meteorological Society, 93, 1363-1387, 10.1175/bams-d-11-00180.1, 2012.

Liu, S., Wang, J. X. L., Liang, X.-Z., and Morris, V.: A hybrid approach to improving the skills of seasonal climate outlook at the regional scale, Climate Dynamics, 46, 483-494, 10.1007/s00382-015-2594-1, 2016.

Sillman, S.: The use of NOy, H2O2, and HNO3 as indicators for ozone-NOx-hydrocarbon sensitivity in urban locations, Journal of Geophysical Research-Atmospheres, 100, 14175-14188, 10.1029/94jd02953, 1995.

Sillman, S., He, D., Cardelino, C., and Imhoff, R. E.: The Use of Photochemical Indicators to Evaluate Ozone-NOx-Hydrocarbon Sensitivity: Case Studies from Atlanta, New York, and Los Angeles, J. Air Waste Manage. Assoc., 47, 1030-1040, 10.1080/10962247.1997.11877500, 1997.

Sun, C., and Liang, X. Z.: Improving U.S. extreme precipitation simulation: Sensitivity to physics parameterizations, Climate Dynamics, to be submitted, 2020a.

Sun, C., and Liang, X. Z.: Improving U.S. extreme precipitation simulation: Dependence on cumulus parameterization and underlying mechanism, Climate Dynamics, to be submitted, 2020b.

Zhang, Y., Wen, X. Y., Wang, K., Vijayaraghavan, K., and Jacobson, M. Z.: Probing into regional O-3 and particulate matter pollution in the United States: 2. An examination of formation mechanisms through a process analysis technique and sensitivity study, Journal of Geophysical Research-Atmospheres, 114, 10.1029/2009jd011900, 2009.

---

## Author Comment (AC2) · 12 Feb 2020

Response to Reviewer #3's comments on He et al. 2019 Atmospheric Physics and Chemistry manuscript

We thank the anonymous reviewer for thoroughly reading our manuscript and providing helpful comments and suggestions, which lead a significant improvement of our manuscript. The detailed responses to major point comments are listed below (text in *italic* and black is the reviewer's comments, and the normal text highlighted in blue is our response):

*This manuscript presents a modeling study of the decadal ozone trend in the US. I am impressed by the significance of the results, but there are still several important issues need to be addressed before publication.*

Response: We appreciate the positive comments from the anonymous reviewer, and the manuscript has been revised according to these comments as listed below.
*Line numbers are based on the revised clean version of the manuscript.

*1 How will the results of CWRF-CMAQ differ from WRF-CMAQ? You can also use WRF to simulate decadal climate with long-term reanalysis. Better to include some description of the advantage of CWRF over WRF.*

Response: The CWRF was developed as a Climate extension of the WRF model incorporating numerous improvements in representation of physical processes and integration of external forcings that are crucial to climate scales, including interactions between land-atmosphere-ocean, convection-microphysics and cloud-aerosol-radiation, and system consistency throughout all process modules (Liang et al., 2012; Qiao and Liang, 2015; Chen et al., 2016; Liu et al., 2016; Qiao and Liang, 2016). It is built with a comprehensive ensemble of many alternate mainstream parameterization schemes for each of key physical processes. To better illustrate the advantage of CWRF, we added this short description in Line 127 of the revised manuscript.

*2 Scaling factors were used to get historical emissions. However, this will keep the spatial distribution the same at 2011 level. Why don't you use the information from other historical inventories, such as EDGAR? Could you discuss how this will affect the results?*

Response: This is a good question. The NEI2011 inventory adjusted with the ground and satellite measurements provides the best available anthropogenic emissions to the CONUS, which has also been used in the operational U.S. national air quality forecast. We used the U.S. National Emissions Trends to produce the scaling factors to generate historical emissions. It is not a perfect solution as pointed out by the reviewer that the assumption of the same spatial distribution may not be true. However, this method guaranteed that the domain total emissions would be consistent with the U.S. official emission trends, which we believe are the best available emissions for CONUS. We believe it is more important to provide total CONUS emissions constrained by the trend data than emissions with more detailed geographic distributions, when our modeling system has relatively coarse spatial resolution and integrates over 20 years. To explain our approach, we added the following sentences in Line 166 as "*Emissions of the baseline year are based on EPA NEI2011 inventory which can provide the best available anthropogenic emissions to the CONUS and are currently used in the operational U.S. national air quality forecast. The usage of APET scaling factors can guarantee the domain total emissions are consistent with the U.S. EPA emissions trend, although assuming the same spatial*

*distribution of anthropogenic emissions from year to year may not be realistic. Without a reasonable observation of actual spatiotemporal variations, it is the cost-effective approach as a first-order approximation to simulate long-term U.S. air quality driven by consistent CONUS total anthropogenic emissions that account interannual trends*".

*3 Chemical initial and boundary conditions were obtained from the default concentration profiles built in CMAQ. For long lived chemical species like ozone, long range transport and stratospheric intrusions would be important. If default concentration profiles are set, how to consider the historical changes in sources outside the US?*

Response: We understand that the ICs and BCs are important to the CMAQ performance. To reduce the impacts of ICs, we spin-up the CMAQ model for two weeks before the 5-yr continuous simulations (e.g., two weeks in December 1999 were used to create ICs for 2000-2004 CMAQ simulations). Based on our experiences in CMAQ modeling and the EPA guidance, two weeks' spin-up should be able to eliminate the influences from ICs.

The stratospheric intrusions are important for the ozone pollution in high altitude regions such as Denver, Colorado. However, the regional CMAQ does not include stratospheric chemistry, and our model top level is at 50 hPa. A potential vorticity (PV)-$O_3$ parameterization was developed recently for the hemispheric CMAQ model (Xing et al., 2016), but it is not available for the regional CMAQ version used in this study. We have discussed this shortcoming of our modeling system in the discussion section.

We agree that the long-range transport (LRT) through BCs can play an important role in the regional modeling of U.S. air quality. Our previous studies (He et al., 2016; He et al., 2018) show that the LRT can contribute up to 10% of ozone and $PM_{2.5}$ in the western United States; these numerical simulations are conducted in relatively short period (5 years) under multiple scenarios with fixed and dynamic LBCs. For climate studies, 5-year continuous integration is usually treated as the minimum time period, while longer simulation is preferred to better capture the climate signature. In this project, we designed a 25-yr experiment from 1990 to 2015, so CWRF downscaling can better represent the regional climate of the CONUS. Due to limited computing resources, we chose not conducting 25-yr global CTM simulations to generate dynamic LBCs conditions for CMAQ, but focused on the ozone pollution change within the United States. We understand that our approach could introduce some uncertainties in this study, and added the following sentences in Line 398 of the discussion session, "*So our current modeling system cannot take the historical changes of air pollution outside the United State into account. That is, the effect of long-range transport of air pollutants through model domain boundaries is presumed to be secondary to the long-term trends over the United States.*" and In Line 403 "*With these increased air pollutant transported into the United States, our study may underestimate the impacts of domestic emission reductions to U.S. ozone pollution, especially in the West Coast and the Southwest.*".

*4 O3/NOy ratio was used as the indicator of VOC or NOx limited. The threshold of was adopted (O3/NOy < 15 indicating the VOC-sensitive regime). How to demonstrate this threshold and ratio is proper and accurate or represent the sensitivity. As model usually has difficulty in capturing the concentrations of NOy, the results might be questionable with this assumption.*
Response: We appreciate the reviewer raised this concern, which is also pointed out by the other reviewer. First, we understand that computer models have difficulty to accurately capture the

ambient $NO_y$ concentrations. In this study, because we did not access long-term research grade $NO_y$ observations from EPA, all the analysis using $O_3/NO_y$ ratios as the photochemical indicators is based purely on CMAQ simulations. So we have to rely on results from the previous studies. Sillman explored the concept using photochemical indicators including $O_3/NO_y$ to identify the regime of ozone photochemical production, finding that the link between the ozone production sensitivity and these indicators is largely unaffected by changes in model assumptions, including emission rates of anthropogenic and biogenic species (Sillman, 1995; Sillman et al., 1997). Observations from urban areas of Atlanta, New York, and Los Angeles was compared with modeling results from the Urban Airshed Model at urban scales, and a threshold of 7 was proposed for using $O_3/NO_y$ ratios as the photochemical indictor (Sillman et al., 1997). Zhang et al. (2009) expanded the study to the CONUS with 1-year CMAQ simulations, suggesting a threshold of 15 for $O_3/NO_y$ ratios. Zhang et al. (2009) used previous CMAQ version 4.4 for this 1-yr CONUS simulations of 2001 at a relatively coarse spatial resolution (36 km), which is close to our 30-km CONUS domain, so we adopted their proposed threshold. We agree that the current manuscript lacked the evaluation of this threshold with our modeling system, and we developed the following approach to test it.

We selected hourly $O_3$, $NO_y$, and $NO_x$ concentrations from CMAQ in the afternoon (defined as 12 pm to 4 pm) in 2014, and calculated the $O_3/NO_y$ ratios. Figure 1a shows scatter density of $O_3/NO_y$ ratios vs. $NO_x$ concentrations, which is calculated based on a $100 \times 100$ bins with $NO_x$ from 0-20 ppbv $NO_x$ (i.e., 0.2 ppbv per bin) and 0-100 $O_3/NO_y$ ratios (i.e., 1 per bin). In the afternoon over the CONUS, the ozone production is mainly in high $O_3/NO_y$ ratio (>15) and low $NO_x$ (less than 2 ppbv) environment, i.e., in the $NO_x$-sensitive regions by thresholds proposed by both Sillman et al. (1997) and Zhang et al. (2009). Figure 1b shows the same density plot, but the color stands for mean $O_3$ concentrations. Both low and high ozone concentrations exist in high $NO_x$ region ($NO_x$ > 4 ppbv), which are usually urban or suburban. Then we calculated the weighted ozone concentrations that equals to the product of $O_3/NO_y$ and $NO_x$ scatter density (Fig. 1a) and mean $O_3$ concentrations (Fig. 1b), which stands for the $O_3$ sensitivity with respect to $O_3/NO_y$ ratios and $NO_y$ concentrations over the CONUS (Fig. 1c). At the national scale, when the weighted ozone concentrations increase with CMAQ $NO_x$ levels, the photochemical production is $NO_x$-sensitive. The region with $O_3/NO_y$ higher than 7 and 11 both have this characteristics, while due to low probability (Fig. 1a) and urban environment (Fig. 1b) we believe the $O_3/NO_y$ threshold of 7 stands for the urban environment. Thus, the $O_3/NO_y$ ratio threshold of 15 is more proper for the CONUS scale analysis. This analysis qualitatively supports our application of results from Zhang et al. (2009).

In summary, due to limited resources and experiment design, identifying a threshold of $O_3/NO_y$ ratio is beyond the scope of this study. Using results from our CMAQ model, we proved that the threshold of 15 should be practical for our study. We added the following sentences in Line 323 as "*The usage of $O_3/NO_y$ ratio was first proposed by Sillman (Sillman, 1995; Sillman et al., 1997). Sillman et al. (Sillman et al., 1997) conducted a case study of observations in urban areas (Atlanta, New York, and Los Angeles) and modeling results from the Urban Airshed Model and suggested the threshold of 7 as the transition region from VOC-sensitive environment to $NO_x$-sensitive environment. Zhang et al. (2009a; 2009b) expanded this method to the CONUS with 1-year observations and CMAQ simulations (36-km spatial resolution) and suggested a threshold of 15 for ozone pollution at the national scale. In this study, we did not have access to the long-term research grade $NO_y$ observations from the AQS network and did not conduct sensitivity experiments (due to computational resource limit) with reduced $NO_x$ emissions following Sillman*

*et al. (1997), so we have to reply on the O₃/NOᵧ threshold from literature. We conducted a simple evaluation of our CMAQ results and found the threshold of 7 could be more proper for urban areas and the threshold of 15 should be more applicable for our study of the whole United State (Figure S1 in the supplementary material). Please note that the O₃/NOᵧ ratio could depend on the modeling framework, so due to the similarity of our modeling system (30-km CMAQ) and the model used in Zhang et al. (2009a; 2009b), our analysis suggests the similar threshold of 15"* and the discussion above to the supplementary material.

a)

[Figure]

b)

[Figure]

c)

[Figure]

Figure 1. Afternoon $O_3/NO_y$ ratios vs. $NO_x$ concentrations simulated by CMAQ in 2014. a) Scatter density, the color contour stands for the probability for each bin; b) $O_3$ concentrations, the color contour stands for the mean $O_3$ over the bins; c) Weighted $O_3$ concentrations. Two black lines stand for the $O_3/NO_y$ ratios of 7 and 11.

Refence:

Chen, L. G., Liang, X. Z., DeWitt, D., Samel, A. N., and Wang, J. X. L.: Simulation of seasonal US precipitation and temperature by the nested CWRF-ECHAM system, Climate Dynamics, 46, 879-896, 10.1007/s00382-015-2619-9, 2016.

He, H., Liang, X.-Z., Lei, H., and Wuebbles, D. J.: Future U.S. ozone projections dependence on regional emissions, climate change, long-range transport and differences in modeling design, Atmospheric Environment, 128, 124-133, https://doi.org/10.1016/j.atmosenv.2015.12.064, 2016.

He, H., Liang, X. Z., and Wuebbles, D. J.: Effects of emissions change, climate change and long-range transport on regional modeling of future US particulate matter pollution and speciation, Atmospheric Environment, 179, 166-176, 10.1016/j.atmosenv.2018.02.020, 2018.

Liang, X.-Z., Xu, M., Yuan, X., Ling, T., Choi, H. I., Zhang, F., Chen, L., Liu, S., Su, S., Qiao, F., He, Y., Wang, J. X. L., Kunkel, K. E., Gao, W., Joseph, E., Morris, V., Yu, T.-W., Dudhia, J., and Michalakes, J.: Regional Climate-Weather Research and Forecasting Model, Bulletin of the American Meteorological Society, 93, 1363-1387, 10.1175/bams-d-11-00180.1, 2012.

Liu, S., Wang, J. X. L., Liang, X.-Z., and Morris, V.: A hybrid approach to improving the skills of seasonal climate outlook at the regional scale, Climate Dynamics, 46, 483-494, 10.1007/s00382-015-2594-1, 2016.

Qiao, F. X., and Liang, X. Z.: Effects of cumulus parameterizations on predictions of summer flood in the Central United States, Climate Dynamics, 45, 727-744, 10.1007/s00382-014-2301-7, 2015.

Qiao, F. X., and Liang, X. Z.: Effects of cumulus parameterization closures on simulations of summer precipitation over the United States coastal oceans, J. Adv. Model. Earth Syst., 8, 764-785, 10.1002/2015ms000621, 2016.

Sillman, S.: The use of NOy, H2O2, and HNO3 as indicators for ozone-NOx-hydrocarbon sensitivity in urban locations, Journal of Geophysical Research-Atmospheres, 100, 14175-14188, 10.1029/94jd02953, 1995.

Sillman, S., He, D., Cardelino, C., and Imhoff, R. E.: The Use of Photochemical Indicators to Evaluate Ozone-NOx-Hydrocarbon Sensitivity: Case Studies from Atlanta, New York, and Los Angeles, J. Air Waste Manage. Assoc., 47, 1030-1040, 10.1080/10962247.1997.11877500, 1997.

Xing, J., Mathur, R., Pleim, J., Hogrefe, C., Wang, J., Gan, C. M., Sarwar, G., Wong, D. C., and McKeen, S.: Representing the effects of stratosphere–troposphere exchange on 3-D O3 distributions in chemistry transport models using a potential vorticity-based parameterization, Atmos. Chem. Phys., 16, 10865-10877, 10.5194/acp-16-10865-2016, 2016.

Zhang, Y., Wen, X. Y., Wang, K., Vijayaraghavan, K., and Jacobson, M. Z.: Probing into regional O-3 and particulate matter pollution in the United States: 2. An examination of formation mechanisms through a process analysis technique and sensitivity study, Journal of Geophysical Research-Atmospheres, 114, 10.1029/2009jd011900, 2009.